# Classical and quantum harmonic mean-field models coupled intensively and extensively with external baths

**Francesco Andreucci**[1,2,*], **Stefano Lepri**[3,4,†], **Stefano Ruffo**[1,2,3,‡], **Andrea Trombettoni**[5,1,2,6,#]

[1]SISSA, Via Bonomea 265, I-34136 Trieste, Italy
[2]INFN Sezione di Trieste, I-34100 Trieste, Italy
[3]Consiglio Nazionale delle Ricerche, Istituto dei Sistemi Complessi, Via Madonna del Piano 10, I-50019 Sesto Fiorentino, Italy
[4]INFN Sezione di Firenze, I-50019 Sesto Fiorentino, Italy
[5]Department of Physics, University of Trieste, Strada Costiera 11, I-34151 Trieste, Italy
[6]CNR-IOM DEMOCRITOS Simulation Center, Via Bonomea 265, I-34136 Trieste, Italy

**Abstract.** We study the nonequilibrium steady-state of a fully-coupled network of $N$ harmonic oscillators, interacting with two thermal reservoirs. Given the long-range nature of the couplings, we consider two setups: one in which the number of particles coupled to the baths is fixed (intensive coupling) and one in which it is proportional to the size $N$ (extensive coupling). In both cases, we compute analytically the heat fluxes and the kinetic temperature distributions using the nonequilibrium Green's function approach, both in the classical and quantum regimes. In the large $N$ limit, we derive the asymptotic expressions of both quantities as a function of $N$ and the temperature difference between the baths. We discuss a peculiar feature of the model, namely that the bulk temperature vanishes in the thermodynamic limit, due to a decoupling of the dynamics of the inner part of the system from the baths. At variance with usual cases, this implies that the steady state depends on the initial state of the particles in the bulk. We also show that quantum effects are relevant only below a characteristic temperature that vanishes as $1/N$. In the quantum low-temperature regime the energy flux is proportional to the universal quantum of thermal conductance.

Email:
*fandreuc@sissa.it
†stefano.lepri@isc.cnr.it
‡ruffo@sissa.it
#andreatr@sissa.it

## 1. Introduction

In the recent years we witnessed a growing interest in the out-of-equilibrium properties of classical and quantum many-body systems, motivated by the remarkable progresses in the the research on nanosized structures and cold atoms or ions physics. From the fundamental point of view, this research theme is part of the aims of nonequilibrium statistical mechanics to justify transport properties from microscopic interactions. A key quantity to understand the transport properties of a system is the thermal conductivity $\kappa$, which is defined by Fourier law:

$$\mathcal{J} = -\kappa \nabla T, \tag{1}$$

where $\nabla T$ is the temperature gradient and $\mathcal{J}$ is the heat flux flowing through the system. For normal (diffusive) systems thermal transport follows the above law, that is, $\kappa$ in an intensive quantity that does not depend on the size of the system.

At the microscopic level, it is well known that the harmonic approximation of interparticle forces yields a violation of Eq. (1): transport is ballistic since quasi-particles travel undisturbed throughout the system bulk. This was demonstrated by Rieder, Lebowitz and Lieb [1] in their seminal study of the non-equilibrium steady state of the simplest example: the harmonic chain with nearest-neighbor coupling in contact with two external reservoirs. They found out that the thermal conductivity scales as $\kappa \sim N$, where $N$ is the number of particles in the chain, and the temperature profile in the bulk of the system is flat, whereas Fourier's law would yield a linear profile. Many later works considered variants of the harmonic crystal, Since in this case one has to deal with linear problems, one can exploit a variety of established techniques like the approach based on transmission coefficients or non-equilibrium Green functions, see the dedicated sections in the reviews [2, 3] and also [4] for a more recent account. Quantum harmonic lattices outside equilibrium have also been studied in various contexts, starting from the simplest case of the one-dimensional chain [5], see e.g. [6] and [7–9].

Besides this, it is also established that Fourier's may be violated in lower dimensions ($d = 1, 2$) also in presence of nonlinear interactions. This is the phenomenon of anomalous (superdiffusive) heat transport that has thoroughly studied in the last two decades [2, 3, 10, 11] in one and two-dimensional, non-linear, non-integrable, momentum-conserving and short-range models. In one dimension, the thermal conductivity may scale as $\kappa \sim N^\alpha$. There are strong evidences (both numerical and theoretical) that the exponent $\alpha$ can be used to categorize different universality classes [11]. In those cases, Eq. (1) should be replaced by its fractional version, yielding nonlinear temperature profiles across the system [12, 13]. Finally, nonlinear integrable models like for instance the Toda chain [14] should generically display ballistic transport mediated by solitons (see [13, 15] and references therein for some recents results). However, there may be exceptions to this intuition, as in the cases of hard-point particles [16] and the classically integrable Landau-Lifshitz spin chain [17].

One can then wonder how, and if, does the picture change for long-range interacting

systems, that is, systems in which the interparticle interaction is of the form $V(r) \sim r^{-d-\sigma}$, in dimension $d$. There is a vast literature regarding the physics of these systems, see for example the reviews [18] and [19], for classical and quantum systems, respectively. At equilibrium they are characterized by non-additivity and critical exponents depending on the value $\sigma$. In particular the critical exponents for $\sigma < 0$ are the mean-field ones obtained by putting $\sigma = -d$, where $d$ is the dimensionality of the system. At nonequilibrium, the dynamics of long-range systems presents metastable states whose lifetime scales as $N$ [20–22] and even lack of thermalization upon interaction with a single external bath [23].

Moreover, since perturbations may propagate with infinite velocities [24, 25] this is expected to yield anomalous transport properties. Indeed, numerical studies of energy transport in classical long-range interacting systems support this expectation [26–31] The long-range version of the the XY model [27, 30] and the Fermi-Pasta-Ulam-Tsingou chain and were analyzed in [28, 30, 32, 33], and violations of Fourier law were observed depending on the exponent $\sigma$ of the interaction. On the quantum side, transport in a mean-field harmonic model was studied in [34]; we also refer once again to [19] and references therein for further results. From an analytical perspective, a classical model with stochastic momentum exchange was studied in [35], while a hydrodynamic approach for transport in long-range quantum magnets was developed in [36] and [37].

In the present work, we study one of simplest cases of long-range interacting non-equilibrium, quantum many-body system: a network of mean-field coupled harmonic oscillators in contact with two external heat reservoirs, modeled large ensembles of Bosonic oscillators at different temperatures. The goal of this paper is two-fold. On one hand, we want to have an analytically treatable model for which one can study both classical and quantum regimes. On the other hand, in presence of long-range interactions a natural question is how to couple the system with the baths. In fact, while in short-range systems the coupling concerns only the boundary particles, in the case of long-range interactions, one could ask what is the effect of the interaction between the bulk of the system and the bulk of the baths. Since within our models we can study both intensive and extensive (clarified below) couplings of the system with the external baths, they provide a useful playground to investigate such question. To be specific, we will consider these baths to be Ohmic and coupled to the system in two different ways:

1) We attach each baths to a finite subset of particles, for instance just one as it is usually done for short-range systems. We will refer to this case as intensive coupling. This case also includes the situation in which a finite number of sites is coupled to the two baths.

2) Since the system is long-range, it also makes sense to ask what happens if we couple an extensive fraction of the sites of the system to the baths, since the interaction between the system and the environment may well be itself long-ranged. Therefore, we will connect both baths to a macroscopic number (i.e., scaling with $N$) of sites. We will refer to this case as extensive coupling.

In Sec. 2 we introduce the model, and the tecniques we are going to use to compute

the flux and the temperature profile. In Sec.3, we study the classical case with intensive couplings, and we discuss the peculiarities of the temperature distribution, while in Sec. 4 we tackle the quantum case. In Sec. 5 and 6 we turn to the case of extensive couplings, in the classical and quantum regime, respectively. Finally we review our results in the different cases, discuss a physical interpretation of our results and draw our conclusions.

## 2. The model and the coupling to external baths

We consider a Hamiltonian system describing a network of $N$ fully-connected harmonic oscillators with displacements $x_i$. The Hamiltonian of the model is:

$$H = \frac{1}{2m} \sum_{i=1}^{N} p_i^2 + \frac{k}{2N} \sum_{ij} (x_i - x_j)^2, \tag{2}$$

where $p_i = m\dot{x}_i$, $k$ is the coupling constant and $N$ is the Kac factor, introduced to make the energy extensive. The Hamiltonian can also be cast in the following form, which we will use throughout the paper:

$$H = \frac{1}{2m} \sum_{i=1}^{N} p_i^2 + \frac{1}{2} \sum_{i,j=1}^{N} \Phi_{ij} x_i x_j, \tag{3}$$

where the matrix $\Phi$ is given by:

$$\Phi_{ij} = 2k \left( \delta_{ij} - \frac{1}{N} \right). \tag{4}$$

We connect the system to two external heat reservoirs at different temperatures, $T_L$ and $T_R$. The subscripts here would refer to "left" and "right" bath, but of course, in the mean-field model there is no notion of spatial ordering. Nonetheless, in order to fix the notation we will keep this nomenclature throughout. In the more general situation in which a subset of $N_L$ sites is coupled to the left bath, at temperature $T_L$, while another subset of $N_R$ sites is coupled to the right bath, at temperature $T_R$. We will also assume that these two baths are two ensembles of Bosonic harmonic oscillators, linearly coupled to the system, as explained in [6].

To study the transport properties of this system we will use the Green's function method, which we will briefly illustrate below (for a full account see [6]). By integrating out the oscillators, we get a set of quantum Langevin equation of motion. Since we are interested in the properties of the stationary state, we then switch to Fourier space. The final result is:

$$-m\omega^2 x_i(\omega) = -\Phi_{ij} x_j(\omega) + (\Gamma_{L,ij}(\omega) + \Gamma_{R,ij}(\omega)) x_j(\omega) + \xi_{L,i}(\omega) + \xi_{R,i}(\omega), \tag{5}$$

where $\xi(\omega)$, the noises introduced by the baths, and $\Gamma_{L,R}(\omega)$, the Fourier transforms of the memory kernels, are related by the fluctuation-dissipation relation:

$$\langle \xi_{L,i}(\omega) \xi_{L,j}(\omega') \rangle = Im[\Gamma_{L,ij}(\omega)](1 + f(\omega, T_{L,})) \frac{\hbar}{\pi} \delta(\omega + \omega'), \tag{6}$$

and an equivalent relation holds for the right bath. The specific form of the $\Gamma$ matrices depends on the spectral density of the baths. From now on, we will consider the case of Ohmic baths, whose memory kernel is purely imaginary: Also, for simplicity of notation, we conventionally assume to couple to the left-hand bath the sites $i = 1...N_L$ and to the right-hand bath the sites $i = N - N_R + 1...N$,

$$\Gamma_{L,ij}(\omega) = i\gamma\omega\delta_{ij}\delta_{i1}, \qquad \Gamma_{R,ij} = i\gamma\omega\delta_{ij}\delta_{1N}, \tag{7}$$

$$\Gamma_{L,ij}(\omega) = i\gamma\omega\delta_{ij}\sum_{k=1}^{N_L}\delta_{jk}, \qquad \Gamma_{R,ij} = i\gamma\omega\delta_{ij}\sum_{k=N-N_R+1}^{N}\delta_{jk}, \tag{8}$$

Eq. (7) defines the intensive coupling case, whereas (8) refers to the extensive coupling case when both $N_L$ and $N_R$ are taken to be proportional to $N$. By plugging (7) in (6) we recover the familiar fluctuation-dissipation relation, for example the classical limit of (6) becomes:

$$\langle \xi_{L/R,i}(\omega)\xi_{L/R,j}(\omega') \rangle = \frac{\gamma k_B T_{L/R}}{\pi}\delta(\omega + \omega')\delta_{ij}. \tag{9}$$

The solution of the equations of motion (5) can be written as:

$$x_i(\omega) = \sum_j G_{ij}(\omega)(\xi_{L,j} + \xi_{R,j}), \tag{10}$$

where the matrix $G_{ij}(\omega)$ is the Green's function, given by:

$$G_{ij}(\omega) = \left[-m\omega^2\mathbb{I} + \Phi + \Gamma_L(\omega) + \Gamma_R(\omega)\right]_{ij}^{-1} \equiv Z^{-1}(\omega). \tag{11}$$

Once $G_{ij}$ is computed, we can use the noise correlation (6) to compute all the correlators of interest. In the following we will report the relevant formulas focusing on the computation of the Green's function (11).

Before proceeding further, let us mention that another way to compute the correlators is to explicitly solve the Fokker-Planck equation in the stationary regime. Since the system is quadratic, this amounts to compute the covariance matrix of the canonical variables. As explained in [1], the covariances satisfy a Lyapunov equation, which can be solved numerically [38]. Note that this approach can be used also for the case of a long-range interaction with $\sigma > -1$ (see below). We anticipate that in the mean-field case we tested the results of the calculations obtained in the large $N$ limit against the numerical solutions finding an excellent agreement.

## 3. Intensive coupling, classical case

### 3.1. Heat flux

In the classical case the flux is given by:

$$\mathcal{J}_{cl} = \frac{k_B\Delta T}{\pi}\int_{-\infty}^{\infty} d\omega \, \mathrm{Tr}\left[G(\omega)\Gamma_L G^\dagger(\omega)\Gamma_R\right], \tag{12}$$

where $\Delta T = T_L - T_R$. Plugging (7) in (12) we get:

$$\mathcal{J}_{cl}^{int} = \frac{k_B \Delta T}{\pi} \gamma^2 \int_{-\infty}^{\infty} d\omega \, \omega^2 |G_{1N}(\omega)|^2. \tag{13}$$

Now we need to compute the Green's function by inverting the following matrix:

$$Z_{ij}^{int} = \begin{cases} -m\omega^2 - i\gamma\omega + 2k(1 - 1/N), & i = j = 1, N \\ -m\omega^2 + 2k(1 - 1/N), & i = j \neq 1, N \\ -2k/N, & i \neq j \end{cases} \tag{14}$$

In order to do it we can employ the Sherman-Morrison formula [39], that reads as follows. Given a matrix $M$, with known inverse $M^{-1}$, and two vectors $\boldsymbol{u}$ and $\boldsymbol{v}$, the inverse of $A = M + \boldsymbol{u}\boldsymbol{v}^T$ is:

$$A^{-1} = M^{-1} + \frac{A^{-1}\boldsymbol{u}\boldsymbol{v}^T A^{-1}}{1 + \boldsymbol{v}^T A^{-1}\boldsymbol{u}}. \tag{15}$$

It is easy to show that the $Z^{int}$ matrix (14) can be written as $-Z^{int} = D^{int} + \boldsymbol{u}\boldsymbol{u}^T$, with:

$$D_{ij}^{int} = \begin{cases} m\omega^2 + i\gamma\omega - 2k, & i = j = 1, N, \\ m\omega^2 - 2k, & i = j \neq 1, N, , \\ 0, & i \neq j, \end{cases} \tag{16}$$

$$u_i = \sqrt{2k/N} \; \forall i. \tag{17}$$

Using (15) we can then compute the Green's function (11) exactly. In particular we can compute $G_{1N}$ and plug the result in (12) to obtain the heat flux:

$$\mathcal{J}_{cl}^{int} = \frac{k_B \Delta T \sqrt{2k/m}}{2\pi} I_1(k_1, N), \tag{18}$$

where the function $I_1$ is given by the following integral ($y = \omega\sqrt{2k/m}$):

$$I_1(k_1, N) = \frac{2k_1^2}{N^2} \int_{-\infty}^{\infty} \frac{(y^2 - 1)^2}{(y^2 - 1)^2 + k_1^2 y^2} \frac{dy}{y^2(y^2 - 1)^2 + k_1^2(y^2 - \frac{2}{N})^2}, \tag{19}$$

where we introduced the dimensionless coupling constant:

$$k_1^2 = \frac{\gamma^2}{2mk}. \tag{20}$$

Consider now the second factor of the denominator of the integrand in (19), as a polynomial in $s = y^2$:

$$s(s - 1)^2 + k_1^2(s - 2/N)^2. \tag{21}$$

As $N \to \infty$ (21) has a vanishing root:

$$s_0(k_1, N) = -4k_1^2/N^2 + o(N^{-2}). \tag{22}$$

By decomposing the integrand in (19) in partial fractions, it is easy to see that the dominant contribution for large $N$ is the one coming from (22):

$$I_1 = \frac{2k_1^2}{N^2} \int_{-\infty}^{\infty} \frac{dy}{y^2 + 4k_1^2/N^2} = \frac{\pi k_1}{N}, \tag{23}$$

and so the heat flux at leading order in $N^{-1}$ reads as:

$$\mathcal{J}_{cl}^{int} = k_B \Delta T \sqrt{2k/m} \frac{k_1}{2N}. \tag{24}$$

We can compare this analytical prediction with the numerical calculation of the integral $I_1$ as a function of $k_1$ for several values of $N$, which is plotted in fig. 1a. It is clear that the two prediction do not match since $I_1$ has a maximum for some optimal value of the coupling and then goes to zero for large $k_1$‡. To get a better understanding of the problem ,we also plot $N I_1(k_1, N)$ as a function of $k_1$ in figures 1a and 1b for different values of $N$: as $N$ grows, the region of agreement between the predicted scaling with $N$ and dependence on $k_1$ (given by 23) grows as well. This fact can be understood in the following way: as $N$ grows the maximum of $I_1$ moves to the right on the $k_1$ axis, as can be seen from figure 1 so when $N \to \infty$ the maximum is virtually at $k_1 = \infty$ so that only the linear region of $I_1(k_1, N)$ is visible.

*3.2. Temperature profile*

We consider the kinetic definition of the temperature of the $i^{th}$ site:

$$T_i = \frac{m}{k_B} \left\langle \dot{x}_i^2 \right\rangle. \tag{25}$$

The classical velocity-velocity correlator can be expressed in terms of the Green's function as [6]:

$$\left\langle \dot{x}_i^2 \right\rangle = \frac{k_B T_L}{\pi} \int_{-\infty}^{\infty} d\omega \omega \left[ G(\omega) \Gamma_L(\omega) G^\dagger(\omega) \right]_{ii} + (L \to R), \tag{26}$$

and by substituting in (7) we get:

$$\left\langle \dot{x}_i^2 \right\rangle = \frac{\gamma k_B T_L}{\pi} \int_{-\infty}^{\infty} d\omega \omega^2 |G_{i1}(\omega)|^2 + \frac{\gamma k_B T_R}{\pi} \int_{-\infty}^{\infty} d\omega \omega^2 |G_{iN}(\omega)|^2. \tag{27}$$

For the first site, $i = 1$, in the large $N$ limit we get:

$$\left\langle \dot{x}_1^2 \right\rangle = \frac{\gamma k_B T_L}{\pi} \int_{-\infty}^{\infty} \frac{\omega^2}{(m\omega^2 - 2k)^2 + \gamma^2 \omega^2} + O(N^{-1}) = \frac{k_B}{m} T_L + O(N^{-1}),$$

where the term $O(N^{-1})$ is proportional to the heat flux. The same formula also holds for $i = N$, with $L \to R$. For all the other sites, $i \neq 1, N$, the velocity-velocity correlator is given by:

$$\left\langle \dot{x}_i^2 \right\rangle = \frac{k_B}{m} \frac{T_L + T_R}{2\pi} I_2(k_1, k_2, N), \tag{28}$$

‡ It is worth noting that this depedence on the coupling constant is qualitativaly the same as the one in short-range case [1].

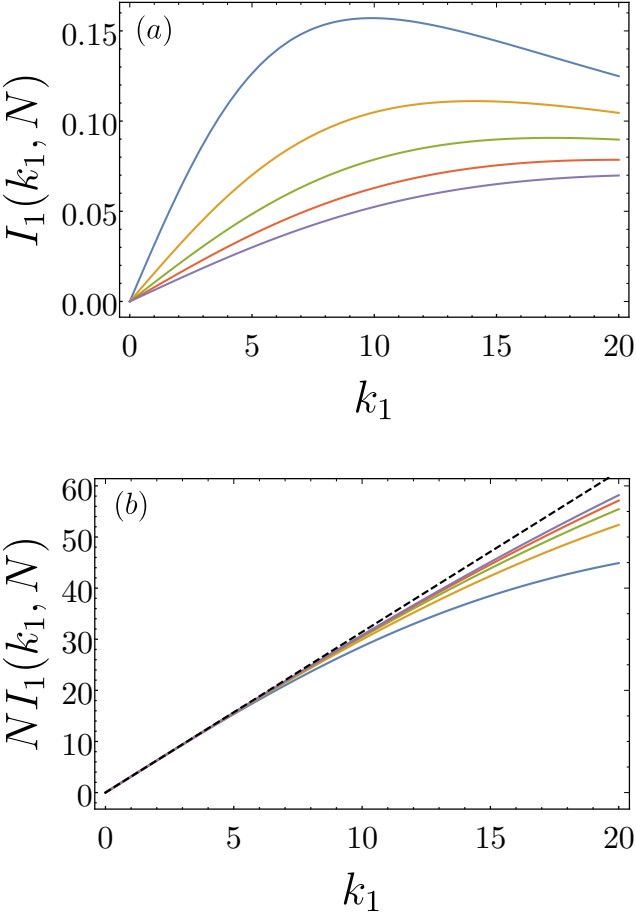

**Figure 1:** In fig. (a) we report the plot of $I_1(k_1, N)$ for $N = 100, 200, 300, 400, 500$ from top to bottom, respectively. Notice how the flux reaches a maximum value and then decreases. In fig. (b) we report the plot of $NI_1(k_1, N)$ for $N = 1000, 2000, 3000, 4000, 5000$, respectively. Note how the agreement with the analytical prediction (represented by the dashed black line) gets better and better as $N$ grows.

with:

$$I_2(k_1, N) = \frac{4k_1}{N^2} \int_0^{+\infty} \frac{dy}{y^2(y^2-1)^2 + k_1^2(y^2 - \frac{2}{N})^2}. \tag{29}$$

This integral can be decomposed in partial fractions and dealt with in the same way as $I_1$: it turns out that at leading order for large $N$ we have $I_2 = I_1/k_1$. Therefore the temperature of the generic $i$-th site is given by:

$$T_{cl,i}^{int} = \frac{T_L + T_R}{2\pi} \frac{I_1}{k_1} = \frac{T_L + T_R}{2N}. \tag{30}$$

This result may appear at first glance unphysical. The temperature of bulk oscillators do not equilibrate to the average of the temperatures of the baths, but rather vanishes in the thermodynamic limit. This is a peculiarity of the model and depends on the choice of the initial conditions. In the next paragraph we give an explanation of this fact based on the analysis of the equation of motion.

*3.3. Analysis of the equations of motion in the stationary regime*

Let's start from the equations of motion in the time domain of the system coupled to the baths: (we set $m = 1$ for simplicity)

$$\ddot{x}_i = -\sum_j \Phi_{ij} x_j + \delta_{i1}(\xi_L - \gamma\dot{x}_i) + \delta_{i1}(\xi_L - \gamma\dot{x}_i), \tag{31}$$

where $\Phi$ is defined in (4), and $\xi_{L,R}$ are Gaussian noises with correlation given by (9). We now introduce the "total magnetization" $M(t) = \sum_i x_i/N$, and $S = x_1 + x_N$. The equations of motion (31) can then be cast in the following form:

$$\ddot{S} = -\lambda S - 2kS + 4kM + \xi, \tag{32}$$
$$\ddot{M} = (\xi - \lambda\dot{S})/N, \tag{33}$$
$$\ddot{x}_i = -2kx_i + 2kM, \quad i = 2, ..., N - 1. \tag{34}$$

Switching to Fourier space, we find the following solution for $M(\omega)$ and the position $x_i(\omega)$ of the uncoupled sites:

$$M(\omega) = \frac{2k - \omega^2}{N\omega^2(\omega^2 + i\lambda\omega - 2k) - 4ik\lambda\omega}\xi(\omega), \tag{35}$$

$$x_i(\omega) = \frac{2kM(\omega)}{2k - \omega^2}. \tag{36}$$

Notice how the pole on the proper frequency of the system $\omega^2 = 2k$ does not give any contribution, as if the baths were unable to properly interact with the system. It is convenient to recast Eq. (36) as:

$$x_i(\omega) = \frac{Q(\omega)}{\omega}\xi(\omega), \quad Q(\omega) \equiv \frac{-2k}{N}\frac{1}{\omega(\omega^2 - 2k) + i\lambda(\omega^2 - 4k/N)}, \tag{37}$$

that has no pole on the dispersion law $\omega^2 = 2k$. The mean square velocity of the $i^{th}$ site then reads as:

$$\langle x_i^2 \rangle = \gamma\frac{T_L + T_R}{\pi}\int d\omega |Q(\omega)|^2, \tag{38}$$

which reproduces exactly formula (28).
To get a better understanding of the physics of the model, let us introduce the relative coordinates $z_i = x_{i+1} - x_i$. Then, equation (34) entails that:

$$\ddot{z}_i = -2kz_i, \quad i = 2, ..., N - 2, \tag{39}$$

so the relative coordinates of the uncoupled particles follow a harmonic motion without being influenced by the baths. This, in turn, means that the initial conditions of the system are essential to determine the properties of the stationary state at long times. Indeed, to solve the equations of motion we should use the Laplace–rather than the Fourier– transform. The use of the latter, made in the previous sections, implicitly assumes $x_i(0) = \dot{x}_i(0) = 0$ for $i = 2, ..., N - 2$. All our results are thus valid, provided we make this assumption on the initial conditions. To support the above considerations in

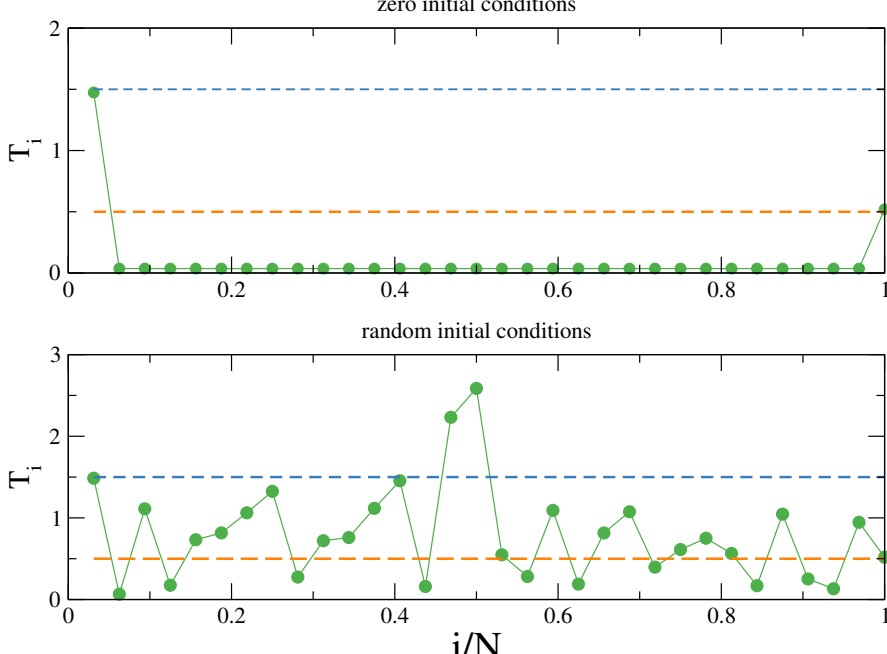

**Figure 2:** Temperature temperature obtained from simulations of the Langevin equation of motion for $N = 32$, intensive baths with $T_L = 1.5, T_R = 0.5, \gamma = 0.5$. Upper panel: zero initial conditions $x_i(0) = \dot{x}_i(0) = 0$ corresponding to the choice adopted in the analytical calculations. Lower panels: random initial conditions where $x_i(0)$ and $\dot{x}_i(0)$ are drawn from a Gaussian distribution with zero average and variance $(T_L + T_R)/2$. Averages are over trajectories of $10^5$ time units.

fig. 2 we report the kinetic temperatures as measured in a Langevin simulation of the equation of motion for two different initial conditions. In the case $x_i(0) = \dot{x}_i(0) = 0$ the results coincide with the result (30). On the other hand, for random initial data the temperatures of the untthermostatted particles remain at their starting value and do not thermalize at all.

We remark that the crucial point is the cancellation of the pole in the dispersion relation of the system, which stems from two properties of the model. The first one is the conservation of the total magnetization $M$ in absence of external baths, which stems from the mean-field nature of the system: from a mathematical point of view, this is related to the $(N-1)$-fold degeneracy of the spectrum of the matrix $\Phi$. The second ingredient is the linearity of the system, that allows the equations of motion to be solved exactly in terms of the Green's function, which in this analysis is given by $Q(\omega)/\omega$. By lifting either of these properties, the temperature profile flattens on the average of the temperatures of the baths.

In order to demonstrate that breaking the degeneracy of of the $\Phi$ may suffice to restore thermalization, we considered two variants of the model. The first one is the quadratic chain with a power-law decaying interaction. The second consist in adding a nearest-neighbors coupling term to (4). The Hamiltonians corresponding to these two

choices are, respectively:

$$H_1 = \sum_i \frac{p_i^2}{2} + \frac{k}{2N_\sigma} \sum_{ij=1}^{N} \frac{(x_i - x_j)^2}{|i - j|^{1+\sigma}}, \quad N_\sigma = \sum_{l=1}^{N} l^{-1-\sigma} \tag{40}$$

$$H_2 = \sum_i \frac{p_i^2}{2} + \frac{k}{2N} \sum_{ij=1}^{N} (x_i - x_j)^2 + \frac{g}{2} \sum_i (x_{i+1} - x_i)^2. \tag{41}$$

Note that in the case of Hamiltonian (40) the expression of the Green's function is not known in the literature, to the best of our knowledge. On the other hand, for the Hamiltonian (41), while it is possible to extend the previous analysis for $g \neq 0$, the calculation does not appear to be straightforward. For these reasons, we decided, in both cases, to solve numerically the Lyapunov equation for the covariance matrix of the models, following the approach of [1]. The results for the temperature profile are plotted in fig. 3. As we can see, in both cases the profile is flat and given by the average temperature of the baths.

Another possibility is to add nonlinear forces. We performed some simulation adding a term $-x_i^3$ to the right-hand side of Eqs.31 [40] and found that kinetic temperatures settle to the the average. Furhermore, the long-range version of the Fermi-Pasta-Ulam-Tsingou chain was numerically studied in [30]: the results for the mean-field case show that the system does thermalize to the average temperature of the baths also in this case.

Finally, we note that one can repeat the analysis presented in this subsection even if the baths induce a coloured noise: once again we obtain that the pole on the proper frequency of the system vanishes.

## 4. Intensive coupling, quantum case

### 4.1. Heat flux

The heat flux in the quantum case is given by [6]:

$$\mathcal{J}_q = \int_{-\infty}^{\infty} d\omega \, \text{Tr} \left[ G(\omega) \Gamma_L(\omega) G^\dagger(\omega) \Gamma_R(\omega) \right] \frac{\hbar\omega}{\pi} \left[ f(\omega, T_L) - f(\omega, T_R) \right]. \tag{42}$$

Futhermore, we will work in the linear response regime:

$$T_L - T_R = \Delta T \ll (T_R + T_L)/2 = T. \tag{43}$$

To get the heat flux for intensive couplings we plug (7) into (42) and we expand to first order in $\Delta T$:

$$\mathcal{J}_q^{int} = \frac{\hbar\gamma^2 \Delta T}{\pi} \int_{-\infty}^{+\infty} d\omega \, \omega^3 |G_{1N}(\omega)|^2 \frac{\partial f(\omega, T)}{\partial T}. \tag{44}$$

The Green function element $G_{1N}$ is the same as in the classical case, and therefore the heat flux can be written as:

$$\mathcal{J}_q^{int} = \frac{k_B \Delta T \sqrt{2k/m}}{4} I_3(k_1, \theta; N), \tag{45}$$

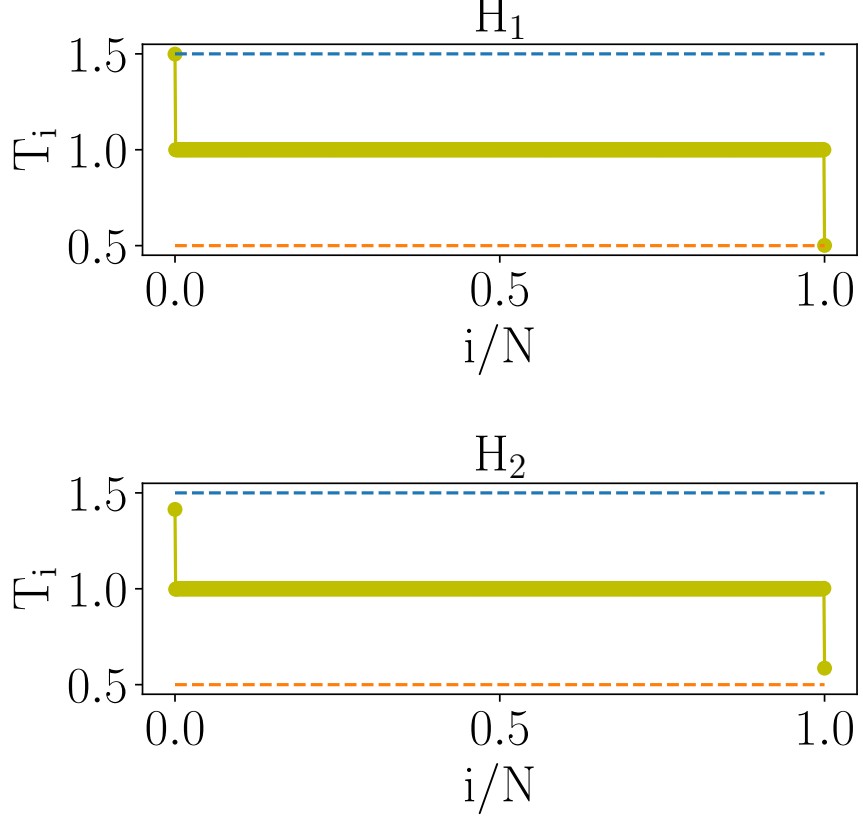

**Figure 3:** Numerical result for the temperature profile $T_{cl,i}^{int}$ for the Hamiltonians (40) and (41) on the top panel and on the bottom panel, respectively. We used the following values for the parameters:$T_L = 1.5$, $T_R = 0.5$ $k = g = 1$, $\sigma = -0.5$.

where we introduced a dimensionless temperature $\theta$ as:

$$\theta = \frac{2k_B T}{\hbar}\sqrt{m/2k}, \tag{46}$$

and the the function $I_3(k_1, \theta, N)$ is given by:

$$I_3 = \frac{4}{\pi}\frac{k_1^2}{\theta^2 N^2}\int_{-\infty}^{\infty} dy \, \frac{(y^2-1)^2}{(y^2-1)^2+k_1^2 y^2}\frac{y^2/\sinh^2(y/\theta)}{y^2(y^2-1)^2+k_1^2(y^2-2/N)^2} \tag{47}$$

The integral $I_3$ (47) is plotted as a function of $\theta$ in fig. 4: as expected, the heat flux goes to zero at low temperatures (when $\theta$ is small), and saturates at high temperatures (when $\theta$ is large). From the figure it is clear that there exist a characteristic temperature scale, which we will call $T_N(k_1)$, that discriminates between the quantum and classical regimes. It turns out that $T_N(k_1)$ goes as $1/N$, as we can see by a direct computation of $I_3$ in the large $N$ limit. This computation is reported in the Appendix, and the final result is:

$$\mathcal{J}_q^{int} = \frac{k_B \Delta T}{2N}k_1\sqrt{\frac{2k}{m}}g\left(\frac{T_N(k_1)}{T}\right), \tag{48}$$

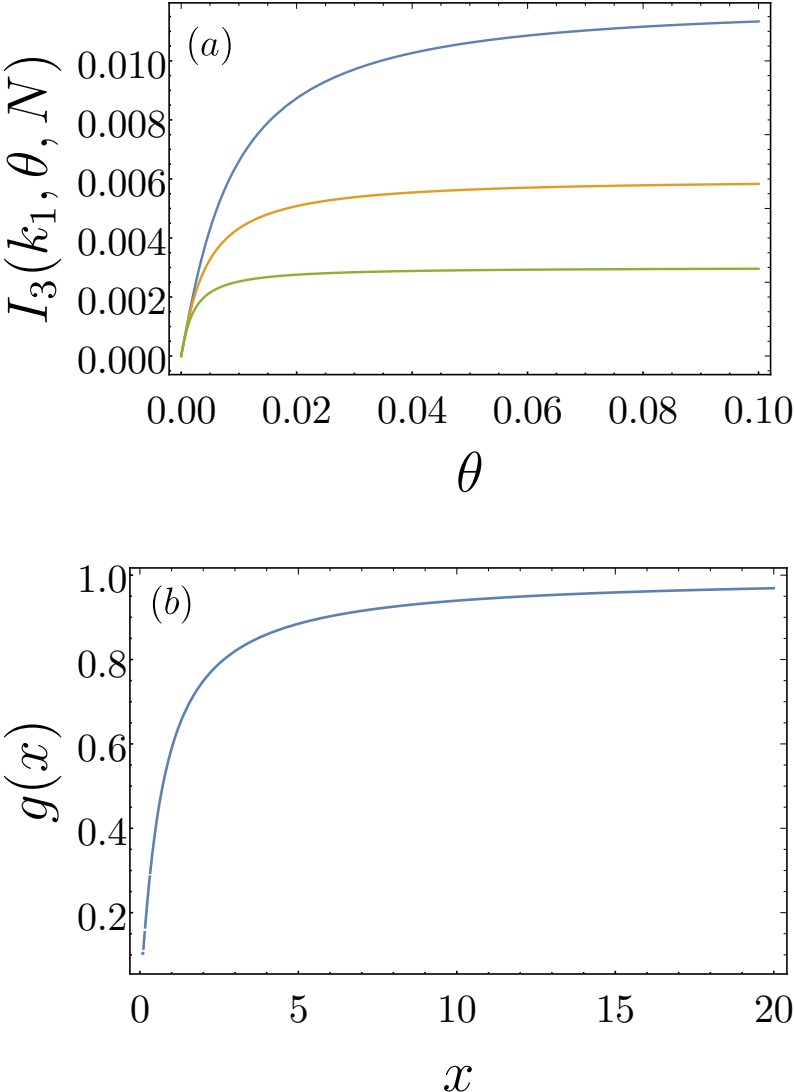

**Figure 4:** In fig. $(a)$ we report $I_3$ as a function of $\theta$, with $k_1 = 3$, and $N = 500, 1000, 2000$ from top to bottom, respectively. In fig. $(b)$ we plot the function $g(x)$ defined in (49).

where the function $g(x)$ is given by:

$$g(x) = \frac{x^2}{\pi^2}\left[\psi^{(1)}\left(1 + \frac{x}{\pi}\right) - \psi^{(1)}\left(1 - \frac{x}{\pi}\right)\right] + \frac{x^2}{\sin^2(x)} - 2x, \tag{49}$$

$$\psi^{(1)}(z) = \frac{d^2}{dz^2}\Gamma(z),$$

being $\Gamma(z)$ the Euler Gamma function. The temperature $T_N(k_1)$ is the intrinsic temperature scale of the system, below which quantum effects are important. It is given by:

$$T_N(k_1) = \frac{k_1}{N}\frac{\hbar}{k_B}\sqrt{\frac{2k}{m}}. \tag{50}$$

To get a better picture of the crossover from the quantum to the classical regime, we consider the ratio between the the quantum (48) and the classical heat flux (24), $\mathcal{J}_q^{int}/\mathcal{J}_{cl}^{int}$.

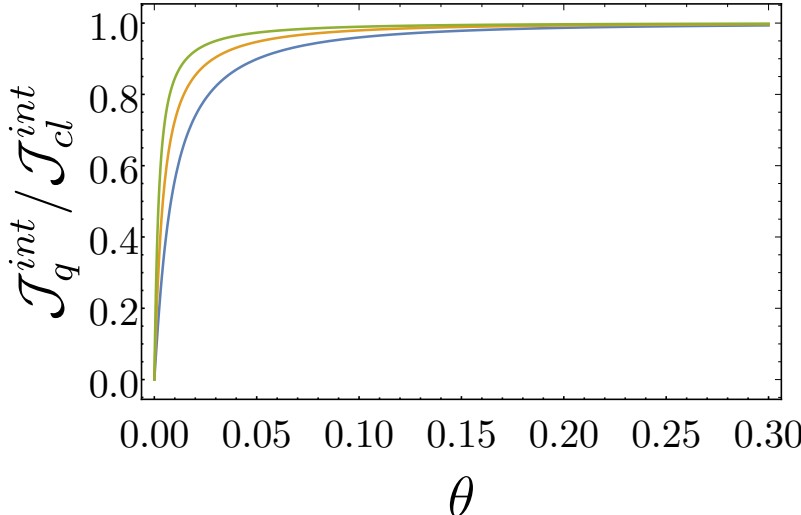

**Figure 5:** We report the plot of the ratio $\mathcal{J}_q^{int}/\mathcal{J}_{cl}^{int}$ as a function of $\theta$ for $k_1 = 3$ and $N = 500, 1000, 2000$ from bottom to top, respectively.

For large $N$, this ratio is given by the function $g(x)$ defined in (49). Its low and high temperature behaviors can be worked out explicitly and are given by:

$$g\left(\frac{T_N}{T}\right) = \begin{cases} 1, & T \gg T_N, \\[2ex] \dfrac{\pi}{3}\dfrac{T}{T_N} \sim TN, & T \ll T_N, \end{cases}$$

where we used the asymptotic formulas for the digamma function. We can see that at high temperature the quantum flux correctly converges to the classical one, while at low temperature it vanishes linearly with $T$. In fig. 5 we plot the aforementioned ratio as a function of $\theta$ for several values of $N$: as $N$ increases, the saturation to the classical value takes place at lower values of $T$. As a final remark, we note that the flux (48) for low temperature is:

$$\mathcal{J}_q^{int} = \left(\frac{\pi^2 k_B^2 T}{3h}\right)\Delta T. \tag{51}$$

Remarkably, the quantity among parentheses is recognized to be the the quantum of thermal conductance, introduced in [41] for heat transport in ballistic quantum wires. It is a universal quantity, independent of all the system parameters (the coupling constants in our case).

*4.2. Temperature profile*

In the quantum case, the velocity-velocity correlator is given by [6]:

$$\langle \dot{x}_i^2 \rangle = \int_{-\infty}^{\infty} d\omega\, \omega^2 \left[\left(G^+(\omega)\Gamma_L(\omega)G^{+\dagger}(\omega)\right)_{ii} \frac{\hbar\omega}{2}\coth\left(\frac{\hbar\omega}{2k_B T_L}\right) + (L \to R)\right], \tag{52}$$

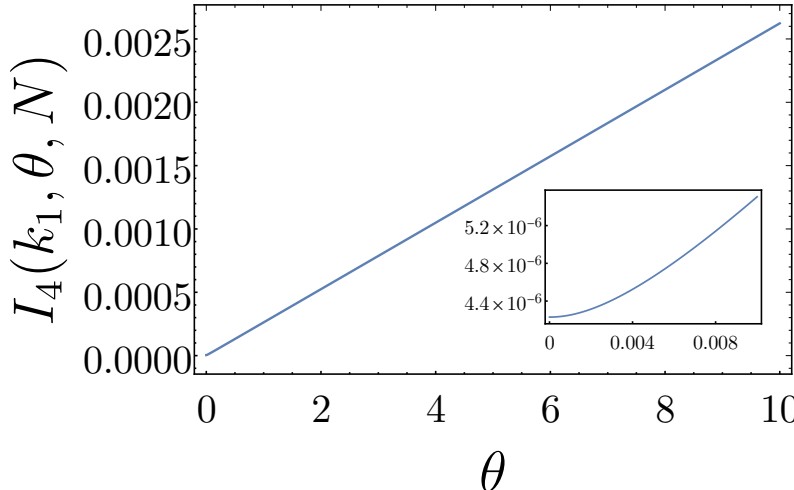

**Figure 6:** Plot of $I_4$ as a function of $\theta$ for $k_1 = 3$ and $N = 1000$. In the inset we plot $I_4$ at low temperatures.

plugging (7) in (52) we get:

$$\langle \dot{x}_i^2 \rangle = \frac{\gamma}{\pi} \int_{-\infty}^{\infty} d\omega \omega^2 \left[ |G_{i1}|^2 \frac{\hbar\omega}{2} \coth\left(\frac{\hbar\omega}{2k_B T_L}\right) + |G_{iN}|^2 \frac{\hbar\omega}{2} \coth\left(\frac{\hbar\omega}{2k_B T_L}\right) \right], \qquad (53)$$

For site number 1, that is, the one coupled to the left bath, the leading order term is:

$$\langle \dot{x}_1^2 \rangle = \frac{\hbar\gamma}{2\pi} \int_{-\infty}^{\infty} \frac{d\omega}{(m\omega^2 - 2k)^2 + \gamma^2\omega^2} \omega^3 \coth\left(\frac{\hbar\omega}{2k_B T_L}\right). \qquad (54)$$

We get the same expression for site $N$, with $L \to R$. The integral in Eq. (54) is logarithmically divergent at large frequencies. This problem is unrelated with the long-range properties of the interaction, it is present even if we couple a single oscillator to two ohmic baths (see for example [42]). The divergence stems from the implicit hypothesis that the bath is able to excite arbitrarily high frequencies, hidden in the choice $\Gamma \sim \gamma\omega$. Physically, there has to be a cutoff at high frequencies.

Now let's consider the case $i \neq 1, N$: in this case it is easy to see that in the linear response regime the term proportional to $\Delta T$ vanishes, and the leading term is of order $\Delta T^0$:

$$\langle \dot{x}_i^2 \rangle = \frac{\hbar\gamma}{2m^2} I_4, \qquad (55)$$

where $I_4$ is given by:

$$I_4 = \frac{1}{N^2} \int_0^{\infty} dy \frac{y \coth(y/\theta)}{y^2(y^2 - 1)^2 + k_1^2(y^2 - 2/N)^2}. \qquad (56)$$

While we cannot solve the integral $I_4$ exactly, we can compute its low-temperature behaviour by approximating the hyperbolic cotangent in (56) with the following series, valid for large $x$:

$$\coth(x) = 1 + 2 \sum_{k=0}^{\infty} e^{-2(k+1)x}. \qquad (57)$$

The integral of the first term can be computed exactly (note that the denominator in (56) is the same as in (19)):

$$I_4(\theta = 0) = \frac{1}{2N^2} \left[ 2\ln\left(\frac{N}{2k_1}\right) + \frac{R}{\sqrt{1-R^2}} \left(\frac{\pi}{2} + \arctan\left(\frac{R}{\sqrt{1-R^2}}\right)\right) \right], \quad (58)$$

where $R$ is the real part of the two roots that do not vanish in the large $N$ limit (which is the same for both of them since they are complex conjugates). The integral of the second term in (57) can be computed fairly easily with some algebra in the large $N$ and low-temperature limit ($T \ll T_N$). In these limits, we find the following result for the temperature profile:

$$T_{q,i}^{int} = \frac{\hbar\gamma k_B}{2m} \left[ I_4(\theta = 0) + \frac{\pi^2}{48} \frac{\theta^2}{k_1^2} \right]. \quad (59)$$

The first term, which is non-zero even if the temperature of the external baths vanishes can be interpreted as a zero point energy of the quantum system.

## 5. Extensive coupling, classical case

We now couple the baths to a number of sites that scales as $N$: the left-hand bath will be coupled to a subset $N_L = \alpha_L N$ of sites starting, while the right-hand bath will be coupled to a subset of $N_R = \alpha_R N$ sites. We will also assume the condition $N_L + N_R \leq N$, that is, a site can be coupled at most with one bath. The computation of the Green's function can be carried out in the same way as in the intensive case, the only difference being that now the $\Gamma$ matrices are given by (8). The matrix that we need to invert is now:

$$Z_{ij}^{ext} = \begin{cases} -m\omega^2 - i\gamma\omega + 2k(1 - 1/N), & i = j = 1...N_L, N - N_R + 1...N, \\ -m\omega^2 + 2k(1 - 1/N), & i = j, \text{ otherwise,} \\ -2k/N, & i \neq j. \end{cases} \quad (60)$$

As in the intensive case, we can decompose this matrix as $-Z^{ext} = D^{est} + uu^T$, where $\boldsymbol{u}$ is defined in (17), and $D^{ext}$ is given by:

$$D_{ij}^{ext} = \begin{cases} m\omega^2 + i\gamma\omega - 2k, & i = j = 1...N_L, N - N_R + 1...N, \\ m\omega^2 - 2k, & i = j, \text{otherwise,} \\ 0, & i \neq j. \end{cases} \quad (61)$$

Therefore the Green's function can be computed exactly also in the case of extensive coupling to the baths.

It is important to note that the results obtained in this section are formally valid also when we couple a finite number of sites, that is, a number that does not scale with $N$, to the baths (a case that has to be considered an *intensive* coupling). An analysis similar to the one of the previous section has to be performed in order to extract the proper dependence on $N$. The result is that the scalings do not change and only the prefactors are affected.

## 5.1. Heat flux

To compute the heat flux, we first substitute (8) in (12) and then we plug in the relevant matrix elements of the Green's function:

$$\mathcal{J}_{cl}^{ext} = \frac{k_B \Delta T}{\pi} \int_{-\infty}^{+\infty} d\omega (\gamma \omega)^2 \sum_{i=N-N_R+1}^{N} \sum_{l=1}^{N_L} |G_{il}|^2 = \frac{k_B \Delta T \sqrt{2k/m}}{2\pi} I_5(k_1, \alpha_L, \alpha_R), \quad (62)$$

where we defined $I_5$ as:

$$I_5 = k_1^2 (2\alpha_L \alpha_R) \int_{-\infty}^{\infty} dy \frac{(y^2-1)^2}{[(y^2-1)^2 + k_1^2 y^2][y^2(y^2-1)^2 + k_1^2(y^2 - (\alpha_R + \alpha_L))^2]}. \quad (63)$$

Note that the sum over the coupled sites collapses to $N_R N_L |G_{il}|^2$ since due to symmetry (and as can be checked by esplicit calculation), $G_{il}$ with $i \neq l$ is actually independent on $i$ and $l$. As a check, we can recover the results (19) of the intensive case by putting $\alpha_R = \alpha_L = 1/N$ in (63).

In fig. 7 we report the plot of $I_5$ as a function of $k_1$ for some fixed values of $\alpha_L$ and $\alpha_R$. The qualitative behaviour is the same as in the intensive case: the heat flux vanishes for both small and strong coupling, and as the fraction of coupled sites decreases, the heat flux decreases as well, as could be expected. It is also interesting to note that in the extensive case the flux does not depend on $N$, in contrast with the $N^{-1}$ scaling of the intensive flux (24). However, as we are going to see in the next section, the temperature profile in the bulk still goes to zero as $N^{-1}$ in the thermodynamic limit for the same reasons as in the intensive case. This seems an inconsistent result, but it is actually only an apparent dichotomy can be reconciled with the following argument. As we saw in section 3 the coupling with the baths is very weak for the particles in the bulk, so we can picture heat transport as heat flowing, at leading order in $N$, directly between the sites that are coupled to the baths. This means that if we increase the energy pumped into the system by a factor $N$–as we do by coupling an extensive number of sites to the baths– the heat flux will increase by that factor, but the temperature profile will still scale as $N^{-1}$.

## 5.2. Temperature profile

The velocity-velocity correlator computed via (26) by substituting (8):

$$\langle \dot{x}_i^2 \rangle = \frac{k_B \gamma}{\pi} \left[ T_L \int_{-\infty}^{\infty} d\omega \omega \sum_{i=1}^{N_L} |G_{ik}(\omega)|^2 + T_R \int_{-\infty}^{\infty} d\omega \omega \sum_{i=N-N_R+1}^{N} |G_{ik}(\omega)|^2 \right]. \quad (64)$$

As in the intensive case, we get different results if $i$ is directly coupled to a bath or not. If $i$ is coupled to the left/right bath we get at leading order $T_{cl,i}^{ext} = T_{L/R}$, as in the intensive case. If $i$ is not coupled to any bath, then we have:

$$T_{cl,i}^{ext} = \frac{\alpha_L T_L + \alpha_R T_R}{\pi N} I_6(k_1, \alpha_L, \alpha_R), \quad (65)$$

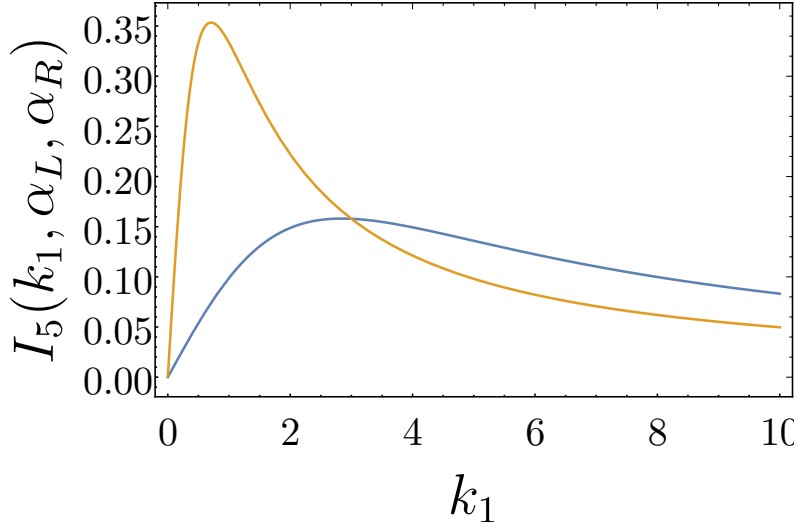

**Figure 7:** Plot of $I_5$ as a function of $k_1$ with $\alpha_L = 1/6, \alpha_R = 1/10$ and $\alpha_L = \alpha_R = 1/2$. The first choice of parameters corresponds to the curve with larger maximum.

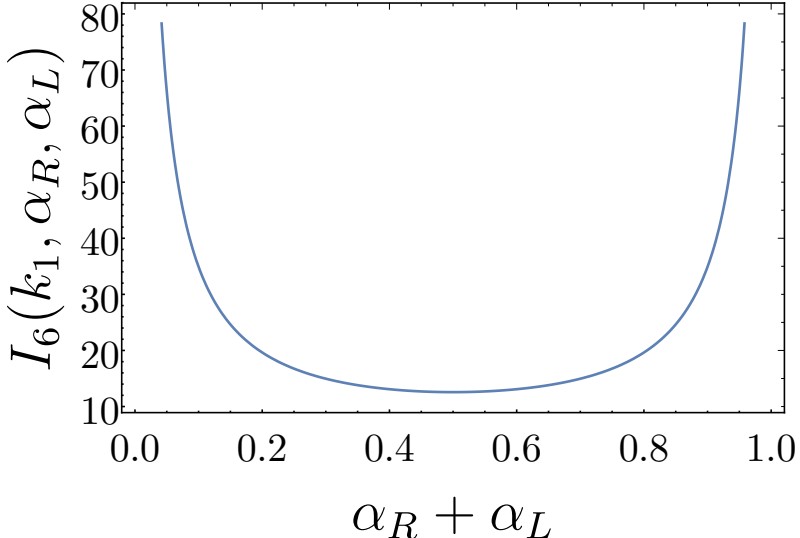

**Figure 8:** The plot of $I_6$ as a function of $\alpha_L + \alpha_R$.

where we introduced the integral $I_6$:

$$I_6 = k_1 \int_{-\infty}^{\infty} dy \frac{1}{[(y^2 - 1)^2 + k_1^2 y^2] \, [y^2(y^2 - 1)^2 + k_1^2(y^2 - \alpha_L - \alpha_R)^2]}, \tag{66}$$

if we set $\alpha_L = \alpha_R = 1/N$ we recover the intensive case (29), apart from a factor of $2N$ due to the different definitions of these integrals. In fig. 8 we report the dependence of $I_6$ on $\alpha_L + \alpha_R$: for $(\alpha_L + \alpha_R) \to 0$ $I_6$ diverges, as it is necessary to match with the intensive case. Also note that even in this case the temperature of the uncoupled sites scales as $N^{-1}$: indeed, the same argument used in section 3 for the intensive coupling case holds also in the extensive case, modulo replacing $S$ with sum of the positions of the sites coupled to

the baths, as can be explicitly checked by solving the equations of motion.

## 6. Extensive coupling, quantum case

### 6.1. Heat flux

The heat flux is obtained by substituing (8) in (42). In the linear response regime we get, as in the intensive case, a factor related to the derivative of the Bose function:

$$\mathcal{J}_q^{ext} = \frac{k_B \Delta T}{\pi} \int_{-\infty}^{+\infty} d\omega (\gamma\omega)^2 \sum_{i=N-N_R+1}^{N} \sum_{l=1}^{N_L} |G_{il}|^2 \frac{\partial f}{\partial T} = \frac{k_B \Delta T \sqrt{2k/m}}{\pi} I_7(k_1, \theta, \alpha_L, \alpha_R),$$

(67)

where we introduced the function $I_7$:

$$I_7 = \alpha_L \alpha_R \frac{k_1^2}{\theta^2} \int_{-\infty}^{\infty} dy \frac{(y^2-1)^2}{(y^2-1)^2 + k_1^2 y^2} \frac{y^2/\sinh(y/\theta)}{y^2 (y^2-1)^2 + k_1^2 (y^2 - (\alpha_L + \alpha_R))^2}.$$

(68)

In fig. 9a we plot $I_7$ as a function of $\theta$ with fixed $\alpha_L, \alpha_R$: as expected, the heat flux vanishes at low temperature, while it saturates at high temperature. Unfortunately, we cannot compute $I_7$ exactly as we did with its intensive counterpart $I_3$ (47), but we can obtain an estimate for the low-temperature behaviour using the following result:

$$\lim_{\theta \to 0} \frac{3}{\pi^2 \theta^3} \frac{y^2}{\sinh^2(y/\theta)} = \delta(y),$$

(69)

so that $I_7$ for small $\theta$ is given by:

$$I_7 = \frac{\pi^2}{3} \frac{\alpha_R \alpha_L}{(\alpha_R + \alpha_L)^2} \theta,$$

(70)

and the heat flux vanishes linearly with the temperature, as in the intensive case (48). In fig. 9b, we report the numerical exact plot of $I_7$ and the low-temperature approximation (70) for several values of $k_1$, respectively. It is evident that the value of $\theta$ below which the linear approximation is valid decreases as a function of $k_1$. This fact implies the presence of a characteristic temperature scale of the system, of which we are however unable to provide an explicit expression, since we are unable to solve (68) exactly. As in the intensive case (4.1), at low temperature the flux can be expressed in terms of the quantum of thermal conductance:

$$\mathcal{J}_q^{ext} = \left( \frac{\pi^2 k_B^2 T}{3h} \right) \frac{4\alpha_R \alpha_L}{(\alpha_R + \alpha_L)^2} \Delta T,$$

(71)

where now we also have a "geometrical factor" that depends on the fraction of coupled sites. Note that for $\alpha_L = \alpha_R$ we recover the intensive result (4.1).

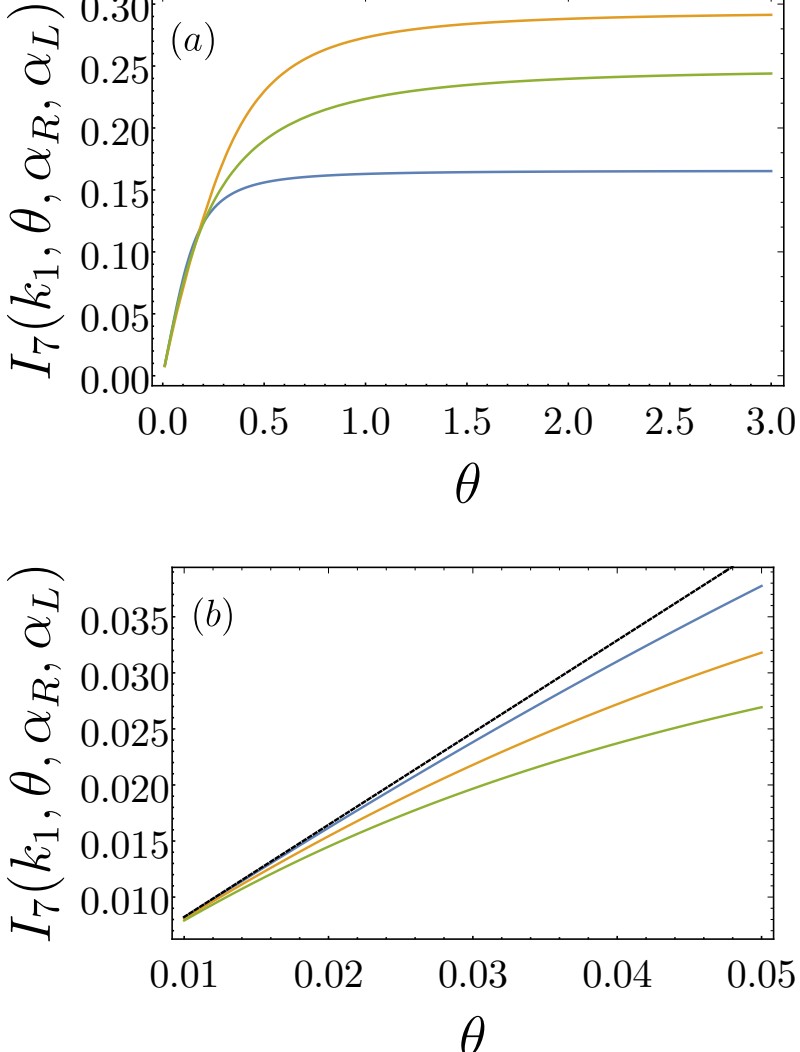

**Figure 9:** In fig. $(a)$ we plot $I_7$ (68) as a function of $\theta$ for $k_1 = 3$, $\alpha_L = 1/20, 1/3, 1/2$ and $\alpha_R = 1/20, 1/5, 1/2$ from bottom to top, respectively. In fig. $(b)$ we plot $I_7$ for low $\theta$ with $\alpha_L = \alpha_R = 1/3$ and $k_1 = 5, 10, 15$ from bottom to top, respectively. The black dashed line is the linear approximation (70).

## 6.2. Temperature profile

In the quantum case we have to plug (8) in (52):

$$\langle \dot{x}_i^2 \rangle = \gamma \left[ \int_{-\infty}^{\infty} d\omega\, \omega \sum_{i=k}^{N_L} |G_{ik}(\omega)|^2 \frac{\hbar\omega}{\pi} \coth\left(\frac{\hbar\omega}{2k_B T_L}\right) \right.$$
$$\left. + \int_{-\infty}^{\infty} d\omega\, \omega \sum_{i=N-N_R+1}^{N} |G_{ik}(\omega)|^2 \frac{\hbar\omega}{\pi} \coth\left(\frac{\hbar\omega}{2k_B T_R}\right) \right]. \tag{72}$$

For the sites directly cupled to the baths , i.e. $i = 1, ..., N_L$ and $i = N - N_R + 1, ..., N$, the correlator diverges due to the contribution of the unphysical high frequencies. On the

other hand, if $i = N_L + 1, ...N - N_R$, then we get:

$$T_{q,i}^{ext} = \frac{\hbar\gamma}{2\pi Nmk_B}\left[\alpha_L I_8(\theta^L, k_1, \alpha_{L,R}) + \alpha_R I_8(\theta^R, k_1, \alpha_L, \alpha_R)\right],\tag{73}$$

where $\theta^{L,R} = 2k_B T_{L,R}/\hbar\sqrt{2k/m}$ the integral $I_8(\theta, k_1, \alpha_L, \alpha_R)$ as:

$$I_8 = \int_{-\infty}^{\infty} dy \frac{y\coth(y/\theta)}{y^2(y^2-1)^2 + k_1^2(y^2 - \alpha_L - \alpha_R)^2}.\tag{74}$$

In the linear response regime we can Taylor expand (73) around $\Delta T = 0$ for $T_{L,R} = T \pm \Delta T/2$:

$$T_{q,i}^{ext} = \frac{\hbar\gamma}{2\pi Nmk_B}\left[(\alpha_L + \alpha_R)I_8^{(0)} + \frac{k_B\Delta T}{\hbar\sqrt{2k/m}}(\alpha_L - \alpha_R)I_8^{(1)}\right],\tag{75}$$

where:

$$I_8^{(0)} = \int_{-\infty}^{\infty} dy \frac{(y^2-1)^2 y\coth(y/\theta)}{[y^2(y^2-1)^2 + k_1^2(y^2 - \alpha_R - \alpha_L)^2]},\tag{76}$$

$$I_8^{(1)} = \theta^{-2}\int_{-\infty}^{\infty} dy \frac{y^2/\sinh^2(y/\theta)}{[y^2(y^2-1)^2 + k_1^2(y^2 - \alpha_R - \alpha_L)^2]}.\tag{77}$$

In fig. 10$a$ we report the plot of $I_8^{(0)}$ and we notice that it goes to a nonzero constant at low temperature: indeed, $I_8^{(0)}(k2 = 0)$ can be interpreted as the contribution to the temperature of the zero-point energy of the particles. We can get an analytical estimate of $I_8^{(1)}$ at low temperature using once again (69):

$$I_8^{(1)} = \frac{\pi^2}{3k_1^2(\alpha_L + \alpha_R)^2}\theta.\tag{78}$$

In fig. 10$b$ we report the plot of $I_8^{(1)}$ and (78) and we see that there is good agreement. Note that (75) entails that, if we couple the same fraction of sites to the left and to the right bath, the term linear in $\Delta T$ vanishes. This was the case for (59), the intensive counterpart of (75), and can be obtained from (75) setting $\alpha_L = \alpha_R = 1/N$. Moreover, since one can check numerically that $I_8^{(0)}$ goes as $\theta^2$ for small $\theta$, we conclude that at low temperature:

$$T_{q,i}^{ext} = \frac{\hbar\gamma}{2\pi Nmk_B}\left[(\alpha_L + \alpha_R)I_8^{(0)}(\theta = 0) + \frac{k_B\Delta T}{\hbar\sqrt{2k/m}}\frac{\pi^2}{3k_1^2}\frac{(\alpha_L - \alpha_R)}{(\alpha_L + \alpha_R)^2}\theta\right].\tag{79}$$

## 7. Conclusions

In this work we analyzed a harmonic mean-field model in various settings. To summarize our results we refer the reader to the table 1, in which we report the scaling of the temperature profile and the heat flux. We considered both the case in which only two sites

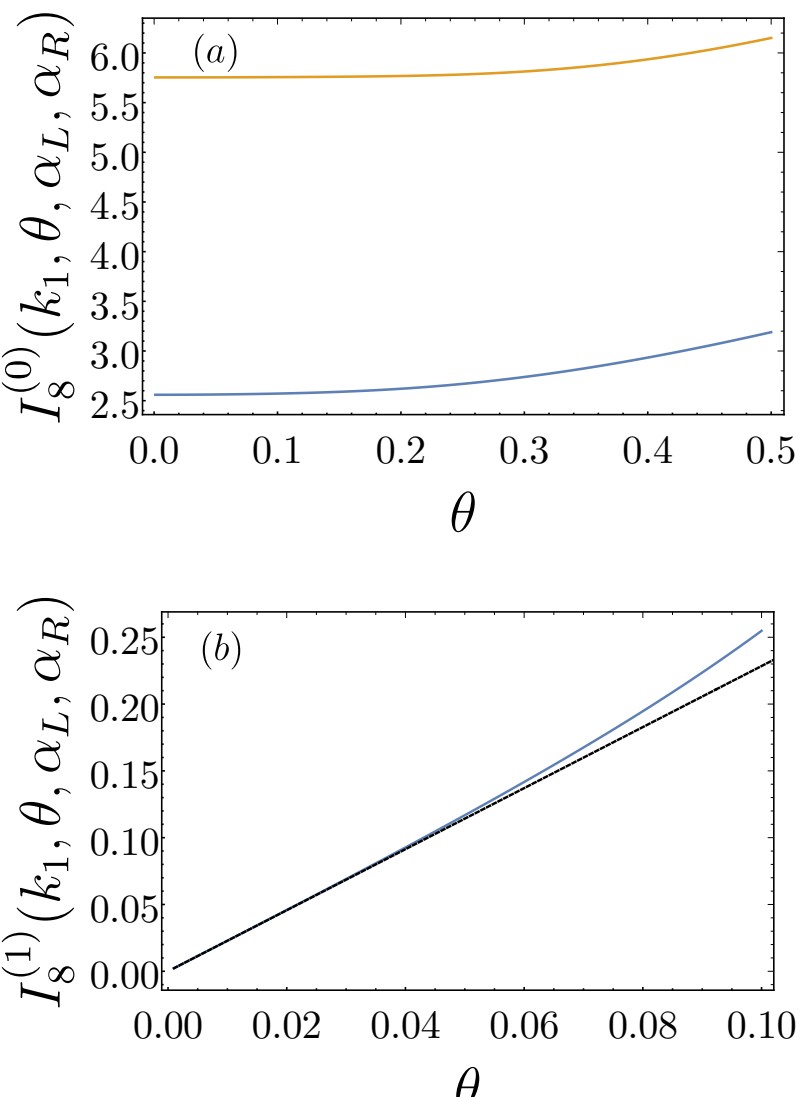

**Figure 10:** In fig. $(a)$ we report the plot of $I_8$ as a function of $\theta$ with $k_1 = 3$ and $\alpha_L + \alpha_R = 0.4, 0.8$ from respectively. In fig. $(b)$ we report the plot of $I_8$ as a function of $\theta$ with $k_1 = 3$ and $\alpha_L + \alpha_R = 0.4$. The black dashed line is the linear approximation (78).

| Coupling | Quantity | Classical | Quantum |
|---|---|---|---|
| Intensive | $\mathcal{J}^{int}$ | $\mathcal{J}_{cl}^{int} \sim \Delta T/N$ | $\mathcal{J}_{q}^{int} \sim T\Delta T$ |
| | $T_i^{int}$ | $T_{cl,i}^{int} \sim T/N$ | $T_{q,i}^{int} \sim T^2 + O(\ln N/N^2)$ |
| Extensive | $\mathcal{J}^{ext}$ | $\mathcal{J}_{cl}^{ext} \sim \Delta T N^0$ | $\mathcal{J}_{q}^{ext} \sim T\Delta T$ |
| | $T_i^{ext}$ | $T_{cl,i}^{ext} \sim N^{-1}$ | $T_{q,i}^{ext} \sim const/N + T/N$ |

**Table 1:** Summary of main results about thermal transport in the mean-field harmonic model (3). We only report the low-temperature behaviour of the quantum results.

are coupled to the baths (which we call intensive coupling case), and the one in which an extensive number of sites is coupled to the baths (which we call extensive coupling case).

Let us now comment on our results, starting from the intensive case. In the classical regime, the peculiar scaling of the temperature profile with $N$ is the result of two properties of the model: having a quadratic Hamiltonian and a degenerate matrix of interactions $\Phi$. If we remove either of these properties, the profile flattens on the average temperature $T$ as we show in figure 3. At low temperatures quantum effects become relevant: interestingly enough, since the scale of temperature of the system scales as $T_N(k_1) \sim N^{-1}$, the region where quantum effects are noticeable shrinks to a point in the thermodynamic limit. At variance, the classical heat flux scales as $N^{-1}$ in the classical regime (in contrast with the short-range case, where the flux is constant for large $N$).

Let us now turn to the extensive coupling case starting from the classical regime. The temperature profile scales once again as $N^{-1}$, but the flux is independent of $N$. Indeed, we are pumping more energy into the system, and so the heat flux is larger. The fact that the temperature profile still scales as $N^{-1}$ is due to the fact that, since all sites are coupled irrespectively of their distance, heat can simply flow from a site coupled to the left bath to one coupled to the right bath. This minimizes, in the large $N$ limit, the amount of energy given to the uncoupled sites.

We also note that at low temperatures, both in the intensive and extensive coupling case, the heat flux vanishes linearly with $T$. The prefactor is given by the quantum of thermal conductance in the intensive case, as expected from [41], and in the extensive case we get a contribution related to the fraction of coupled sites.

It would be interesting to futher study the quantum regime in the extensive coupling case to better understand the dependence of the temperature scale of the system with respect to the coupling constant to the baths and the fractions of coupled sites. From the analysis conducted in this paper, one concludes that for a mean-field system the coupling to an external bath essentially affects only the sites directly coupled to the bath. It would be interesting to see if and to what extent this property stays true if the role of the bath is played by a subsystem that we trace out, for example, in the computation of the entanglement entropy of the system.

*Note added:* During the completion of this manuscript, an interesting and related paper by L. Defaveri, C. Olivares and C. Anteneodo [43] appeared in the arXiv. The authors study heat transport in the same model, for the classical case with intensive couplings, while we also considered the quantum case and the one with extensive coupling. Our results and conclusions are in perfect agreement with theirs in the classical case with intensive coupling and equal masses. They extend their analysis to the case of graded and random masses, where the degeneracy of the model is removed and the system reaches the thermal state. In our paper we show that this also happens if one breaks the degeneracy by adding a power-law long-range interaction or a nearest-neighbor one.

**Acknowledgments**

SL acknowledges support from the program *Collaborations of excellence in research and education* granted by SISSA (Trieste, Italy) where this work has been initiated. This work is part of the MIUR-PRIN2017 project *Coarse-grained description for non-equilibrium systems and transport phenomena (CO-NEST)* No. 201798CZL.

**Appendix: Calculation of $I_3$**

Since the denominator of the integrand in (47) is the same as the one in (19) we can once again exploit the presence of the vanishing root. In the large $N$ limit, the dominant contribution to $I_3$ is:

$$I_3 = \frac{2}{\pi} \frac{k_1 a}{N} \int_{-\infty}^{\infty} \frac{dx}{x^2 + a^2} \frac{x^2}{\sinh^2(x)}, \tag{A.1}$$

where we conveniently made the change of variable $x = y/k_2$ and $a$ is given by:

$$a = \frac{2k_1}{k_2 N} = T_N/T, \quad T_N = \frac{\hbar k_1 \sqrt{2k/m}}{k_B N}. \tag{A.2}$$

As in the classical case (19), we cannot directly take the limit $N \to \infty$, because in this limit $a = 0$ and (A.1) diverges. To compute (A.1) we employ contour integration and

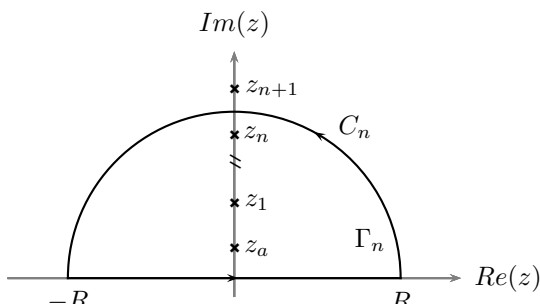

**Figure A1:** The contour $\Gamma_n$ used to compute the integral $I_3$

Cauchy's theorem. Let us introduce the following function of complex variable $z$:

$$f(z) = \frac{z^2}{z^2 + a^2} \frac{1}{\sinh^2(z)}, \tag{A.3}$$

The poles of $f$ are all located on the imaginary axis, at the following positions (with the corresponding residue):

$$z_n = in\pi, \quad n \in \mathbb{Z}\{0\}, \quad \text{Res}_n = \frac{2i\pi a^2 n}{(a^2 - \pi^2 n^2)^2}, \tag{A.4}$$

$$z_{\pm a} = \pm ia, \qquad\qquad \text{Res}_{\pm a} = \mp \frac{ia}{2\sin^2(a)}. \tag{A.5}$$

Consider now the contour $\Gamma_n$ plotted in figure A1: it is composed by a segment $[-R, R]$ and a semicircle $\mathcal{C}_n$ of radius $R$, which is such that $\Gamma_n$ contains the first $n$ $z_k$ poles and the one in $z_a$. Let now be $I_R$ the integral of $f(z)$ over the aforementioned segment. Then, by the residue theorem, we have:

$$I_R = \int_{-R}^{R} dx f(x) = -\int_{\mathcal{C}_n} dz f(z) + 2\pi i \left[ \sum_{k}^{n} \text{Res}_k + \text{Res}_{+a} \right]. \tag{A.6}$$

$I_3$ can then be obtained by taking the limit $R \to \infty$ of $I_R$ as follows:

$$I_3 = \frac{2}{\pi} \frac{k_1 a}{N} \lim_{R \to \infty} I_R. \tag{A.7}$$

We now have to compute the limit $R \to \infty$, which also entails the limit $n \to \infty$ of the right-hand side of (A.6). For large $R$ the integral of $f$ over $\mathcal{C}_n$ is:

$$\int_{\mathcal{C}_n} dz f(z) \approx iR \int_0^{\pi} \frac{e^{i\theta} d\theta}{\sinh^2(Re^{i\theta})} = iR(-2i\coth(R)/R) \to 2. \tag{A.8}$$

The sum over the residues becomes a series that can be resummed. We can thus finally express $I_3$ as:

$$I_3 = \frac{2k_1}{N} g(a), \tag{A.9}$$

where the function $g(x)$ is given by (49).

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
