# Peer review of "Classical and quantum harmonic mean-field models coupled intensively and extensively with external baths"

_SciPost Physics Core_

## Round 1 · Referee Report · Anonymous (Referee 1) · 2022-2-8

Report

In this manuscript, the authors study the temperature profile and the heat flux in a system of fully-connected harmonic oscillators in both classical and quantum regimes. They also consider different types of system-to-bath coupling, referred to as extensive and intensive.

The main results are obtained analytically using the Green functions, with the support of some numerical results, and their properties are analyzed including the effect of degeneracy (and its breaking) in the temperature profile and the scaling of the heat current $J$ with the system size $N$ for both regimes.

Overall, the paper is well written and the results are presented clearly. Heat transport in systems with long-range interactions is an important problem and the analytical results obtained by the authors are relevant and meet the acceptance criteria.

There are only minor points placed in the "Requested changes" section:

Requested changes

1- At the end of section 3.2, the authors mention a numerical result where a non-linear interaction (a quartic potential) is added to break the degeneracy of the system. When the authors state: “numerical simulations”, are they referring to the Langevin equations (as in Fig. 2)? This point should be present in the manuscript,

2- In Fig. 8, the value of $k_1$ is not informed,

3- some typos: at the second paragraph of the introduction there is a "Since" capitalized, in the paragraph about the XY and FPUT models, there is an extra "and" before "were analyzed", in the paragraph beneath Eq. (66) there is a "modulo" that seems out of place.

  • validity: -
  • significance: -
  • originality: -
  • clarity: -
  • formatting: -
  • grammar: -

Author:  Francesco Andreucci  on 2022-04-13  [id 2381]

(in reply to Report 1 on 2022-02-08)

We thank the Referee for the positive evaluation of our manuscript and for their remarks and comments. Below we provide the answer to the points raised by the Referee.

Referee: 1- At the end of section 3.2, the authors mention a numerical result where a non-linear interaction (a quartic potential) is added to break the degeneracy of the system. When the authors state: "numerical simulations" are they referring to the Langevin equations (as in Fig. 2)? This point should be present in the manuscript

Authors: Yes, the simulations were done using the same method that we used to obtain Fig.2. We added a comment on this point in the third paragraph of page 10.

Referee: 2- In Fig. 8, the value of k1 is not informed.

Authors: Thanks, we added the value of k1.

Referee: 3- some typos: at the second paragraph of the introduction there is a "Since" capitalized, in the paragraph about the XY and FPUT models, there is an extra "and" before "were analyzed", in the paragraph beneath Eq. (66) there is a "modulo" that seems out of place.

Authors: Thanks for noticing these typos, we corrected them.

---

## Round 1 · Referee Report · Anonymous (Referee 2) · 2022-2-20

Strengths

The authors have calculated various nonequilibrium steady-state properties of an open, one-dimensional network of harmonic oscillators where each oscillator is connected to all other oscillators. They have applied nonequilibrium Green's function method to find analytical results and simulations to complement in some cases. Overall, the authors have made a detailed analysis to understand and explain their different findings, which are some cases interesting.

Weaknesses

I find the paper has not been written carefully, as I try to point out below. There are too many typos, grammar mistakes, and missing punctuation symbols. I am not able to list here all but a few below.

a) Fourier's law is written in three different ways: Fourier law, Fourier's, and Fourier's law!

b) Please check these sentences in your manuscript!

"Many later works considered variants of the harmonic crystal, Since in this case one has to deal with linear..."

"This is the phenomenon of anomalous (superdiffusive) heat transport that has thoroughly studied in the last two .."

"To get a better understanding of the problem ,we also plot $N I_1(k_1, N )$ as a function of $k_1$ in figures 1a and 1b for different values of N ..."

"On the other hand, for random initial data the temperatures of the untthermostatted particles remain .."

"(and as can be checked by esplicit calculation),..."

Report

I have some queries and suggestions, which I list below.

  1. In the Introduction, the authors write, "This is the phenomenon of anomalous (superdiffusive) heat transport that has thoroughly studied in the last two decades [2, 3, 10, 11] in one and two-dimensional, non-linear, non-integrable, momentum-conserving and short-range models." Do the authors consider normal heat conductions in a chain of coupled rotators (Phys. Rev. Lett. 84, 2144 (2000), Phys. Rev. Lett. 84, 2381 (2000)), double-well type potentials (Phys. Rev. E 86, 041102 (2012)) and asymmetric potentials between two neighboring sites of the chain in the above statement? It might be helpful to add some discussion for these cases there!

  2. Can the authors mention how the classical heat flux scales with system sizes for the models in Eqs. 40 and 41?

  3. I guess the degeneracy of the spectrum of the potential matrix $\Phi$ will be broken if external harmonic pinnings are added at sites $1, N$. Is it going to be the case in your study for getting a flat temperature profile in the middle sites? Will such boundary pinning potentials change the heat flux scaling with $N$ in your studied mean-field model? Such external pinning potentials play an essential role in heat conduction in different ordered and disordered models, e.g., J. Stat. Phys. 131, 535 (2008), Phys. Rev. B 81, 064301 (2010).

  4. I wonder why the authors call their harmonic network "quantum many-body system"! It's undoubtedly not a many-body system of non-trivially (nonlinear) interacting particles for which the phrase "quantum many-body system" is generally used in condensed matter physics.

  5. Can the authors please comment on a higher-dimensional version of the studied model?

Requested changes

  1. The first term in Eq. 22 is an order of $1/N^2$, then what is other $O(N^{-2})$ dropped there?

  2. Since the authors get the classical heat flux in the mean-field model scales with system size as $N^{-1}$ for intensive coupling, Fourier's law is valid here. Since the authors discussed so much on Fourier's law in the Introduction, it would be apt to discuss its validity and break-down in their explored models.

  • validity: good
  • significance: ok
  • originality: good
  • clarity: low
  • formatting: reasonable
  • grammar: below threshold

Author:  Francesco Andreucci  on 2022-04-13  [id 2382]

(in reply to Report 2 on 2022-02-20)

We thank the Referee for the comments. We made a careful reading to eliminate the typos and grammar mistakes present in the previous version. We also addressed the queries and suggestions as detailed in the reply below.

Referee: 1. In the Introduction, the authors write, "This is the phenomenon of anomalous (superdiffusive) heat transport that has thoroughly studied in the last two decades [2, 3, 10, 11] in one and two-dimensional, non-linear, non-integrable, momentum-conserving and short-range models." Do the authors consider normal heat conductions in a chain of coupled rotators (Phys. Rev. Lett. 84, 2144 (2000), Phys. Rev. Lett. 84, 2381 (2000)), double-well type potentials (Phys. Rev. E 86, 041102 (2012)) and asymmetric potentials between two neighboring sites of the chain in the above statement? It might be helpful to add some discussion for these cases there!

Authors: The sentence we wrote needs indeed some further explanation. It is now understood that, for models with three conservation laws, superdiffusive heat transport is generic. The coupled rotors case is different since it only has two conserved quantities. The double-well case should instead belong to the first case but, admittedly, it shows regimes of almost normal transport that are in our view not yet understood. We added a discussion in the introduction to explain the cases mentioned by the Referee.

Referee: 2. Can the authors mention how the classical heat flux scales with system sizes for the models in Eqs. 40 and 41?

Authors: These two cases are currently under investigation, and we are finding somehow unexpected results that should be understood better. In particular, the model in Eq.40, that is, the long-range power-law coupling, does not allow for an analytical solution of the Green's function. Furthermore, the numerical analysis is hindered by the presence of strong finite-size effects, boosted by the tails of the interaction. The other model, mean-field plus a nearest-neighbors interaction, is analytically solvable, but the expression of the Green's function is very involved. We are still trying to extract the asymptotic limit N->infinity. Numerically, we see that if the coupling of the mean-field interaction $k$ is much larger than the one of the short-range interaction $g$ we see the scaling N^-1 as in the mean-field case. On the other hand, for $g$ much larger than $k$, we see that the flux does not depend on N, as in the short-range case. To properly study the cross-over region it would be useful to have at least some hints from the analytical treatment of the problem.

Referee: 3. I guess the degeneracy of the spectrum of the potential matrix Φ will be broken if external harmonic pinnings are added at sites 1,N. Is it going to be the case in your study for getting a flat temperature profile in the middle sites? Will such boundary pinning potentials change the heat flux scaling with N in your studied mean-field model? Such external pinning potentials play an essential role in heat conduction in different ordered and disordered models, e.g., J. Stat. Phys. 131, 535 (2008), Phys. Rev. B 81, 064301 (2010).

Authors: This is a very interesting point, and we thank the Referee for raising it. Indeed, in order to have a non-vanishing temperature in the bulk, the degeneracy of \Phi needs to be macroscopically broken. One can write down the equations of motion for the system with pinned sites 1 and N, and once again the relative positions of the particles are decoupled from the baths, i.e. eq. 39 continues to be true. Other weak breakings of the degeneracy of \Phi, for example a pinning potential for a macroscopically large number of sites, will alter the number of uncoupled degrees of freedom satisfying eq.39, but that number remains of order N. We clarified this point in the paper. Performing the same analysis as done in the manuscript, we can see that the flux scales as N^-1 also in the pinned case.

Referee: 4. I wonder why the authors call their harmonic network "quantum many-body system"! It's undoubtedly not a many-body system of non-trivially (nonlinear) interacting particles for which the phrase "quantum many-body system" is generally used in condensed matter physics.

Authors: We agree with this comment, the phrasing was indeed misleading, and we removed the corresponding line.

Referee: 5. Can the authors please comment on a higher-dimensional version of the studied model?

Authors: Since the model is mean-field, considering a higher-dimensional lattice would not change the Hamiltonian. Indeed, by labelling the sites of the lattice with numbers 1 to N, the potential energy would still be given by the sum in Eq.2 in the main text, where N is the total number of sites of the lattice. In case the Referee was asking about a model with vector particles of s components, in that case the Hamiltonian simply splits in s copies of the model we considered in the paper, so one has only to replace N with s*N.

Referee: 1. The first term in Eq. 22 is an order of 1/N2, then what is other O(N−2) dropped there?

Authors: In Eq.22 we are using the small-o notation, that is o(x) is a quantity that goes to 0 faster than x. We nonetheless added a line below eq.22 to clarify the point.

Referee: 2. Since the authors get the classical heat flux in the mean-field model scales with system size as N−1 for intensive coupling, Fourier's law is valid here. Since the authors discussed so much on Fourier's law in the Introduction, it would be apt to discuss its validity and break-down in their explored models.

Authors: We disagree on this point. Even if the flux scales as N^-1, Fourier's law does not hold here, indeed the temperature profile is not linear, but flat. Microscopically, this is due to the lack of diffusion mechanism in the dynamics.

---

## Round 1 · Referee Report · Anonymous (Referee 3) · 2022-3-2

Report

The authors investigate steady state transport through a mean field harmonic oscillator model, where the non-equilibrium state is driven by coupling to two external heat baths also modeled by harmonic oscillators. This model is exactly solvable by using results from A. Dhar and D. Roy, Journal of Statistical Physics 125 (2006) 801, which allows the authors to derive a number of analytical results for different kinds of coupling to the heat baths and classical vs. quantum oscillators.

My main criticism is the missing motivation for this model in the context of heat transport. General statements like the importance of long-range interactions are not sufficient as a motivation. This manuscript would benefit greatly from an example where transport through a mean field harmonic oscillator (i.e. fully connected harmonic oscillators without spatial structure) is relevant. I cannot think of such an example, therefore I consider the significance of this work to be insufficient. But I would be glad if the authors can prove me wrong on this point.

Requested changes

This manuscript needs an example where transport through a mean field harmonic oscillator (i.e. fully connected harmonic oscillators without spatial structure) is relevant.

  • validity: good
  • significance: poor
  • originality: low
  • clarity: ok
  • formatting: reasonable
  • grammar: acceptable

Author:  Francesco Andreucci  on 2022-04-13  [id 2383]

(in reply to Report 3 on 2022-03-02)

We thank the Referee for the comment, to which we reply below.

Referee:
My main criticism is the missing motivation for this model in the context of heat transport. General statements like the importance of long-range interactions are not sufficient as a motivation. This manuscript would benefit greatly from an example where transport through a mean field harmonic oscillator (i.e. fully connected harmonic oscillators without spatial structure) is relevant. I cannot think of such an example, therefore I consider the significance of this work to be insufficient. But I would be glad if the authors can prove me wrong on this point.

Authors:
On one hand, it is true that the in the fully connected case there is no spatial distance between the constituents of the system. While this is not a problem for the study of equilibrium -- and critical -- properties, the doubt raised by the Referee about transport is certainly very sensible. However, dynamics can be studied as well in mean-field systems, and so transport, e.g. perturbing the system in two points and measuring its response. Whether these transport properties provide the limit \alpha \to 0 of the same system but with an interaction 1/r^\alpha is another question, and in our opinion very interesting per se. For this reason we think mean-field models such the ones studied in this paper have a paradigmatic importance in their own. On the other hand, systems with flat interactions/couplings are, to our knowledge, also connected to experimental setups. In https://arxiv.org/pdf/1702.07653.pdf (section 3 in particular), it is shown that a particular setup of cold atoms in a cavity reproduces the stationary state of the Hamiltonian Mean Field model. The second setup featuring spin systems with flat interactions is in trapped ions: in https://arxiv.org/pdf/2109.01063.pdf (fig 1d pag. 8) and https://www.nature.com/articles/nature10981.pdf. Admittedly, none of these systems are described exactly by the model that we studied in our manuscript, but anyway one can think about variants of them where there is a coupling with another system (or possibly two) acting as a bath, so that one can study dynamics and transport. We added a reference to these systems to further motivate our study. Furthermore, it is perfectly legitimate to imagine an experimental setup in which a big molecule is attached to heat reservoirs, as done in nanoscale experiments (https://www.nature.com/articles/s41586-019-1420-z). The mean-field model would then be the roughest description of the molecule. In this respect it is not necessary to consider a spatial structure along which heat propagates; what matters is the global exchange of heat between the system and the reservoirs.
We conclude then that the mean-field limit is important: 1) as a guide to interpret results for power-law couplings with small, but not vanishing, exponent; and 2) in relation with possible experimental realisations of dynamics and transport of quantum systems with flat interactions/couplings.

---

## Editorial Decision

resubmitted